# Electrophysiology and Fluorescence Spectroscopy Approaches for Evaluating Gamete and Embryo Functionality in Animals and Humans

**DOI:** 10.3390/biom12111685

**Published:** 2022-11-14

**Authors:** Raffaele Boni, Alessandra Gallo, Elisabetta Tosti

**Affiliations:** 1Department of Sciences, University of Basilicata, Via dell’Ateneo Lucano, 10, 85100 Potenza, Italy; 2Department of Biology and Evolution of Marine Organisms, Villa Comunale, 80121 Napoli, Italy

**Keywords:** sperm functionality, oocyte physiology, embryo physiology, electrophysiology, fluorescence spectroscopy

## Abstract

This review has examined two of the techniques most used by our research group for evaluating gamete and embryo functionality in animal species, ranging from marine invertebrates to humans. Electrophysiology has given access to fundamental information on some mechanisms underpinning the biology of reproduction. This technique demonstrates the involvement of ion channels in multiple physiological mechanisms, the achievement of homeostasis conditions, and the triggering of profound metabolic modifications, often functioning as amplification signals of cellular communication. Fluorescence spectrometry using fluorescent probes to mark specific cell structures allows detailed information to be obtained on the functional characteristics of the cell populations examined. The simple and rapid execution of this methodology allowed us to establish a panel helpful in elucidating functional features in living cells in a simultaneous and multi-parameter way in order to acquire overall drafting of gamete and embryo functionality.

## 1. Introduction

In animals, reproductive efficiency depends on many variables affecting the individual functions (genetics, endocrine, immunity, and pathology) and their relationship with the environment (seasonality, thermal stress, and pollutants). In this context, the fertility of the gametes, i.e., their ability to fertilize and produce viable progeny, plays a pivotal role due to the ability of these cells to achieve their intended task, which is procreation. The fertilization success with embryo development and the birth of full-term offspring represent a reliable expression of the gamete fertility potential and the result of the numerous effects interfering with reproductive efficiency. The fertility assessment, however, is not usually practiced due to high costs, time consumption, and organizational difficulties and, in humans, this option is not applicable. Therefore, it was necessary to develop independent in vitro tests capable of estimating gametes’ fertilization and developmental competence with good accuracy. Each of these tests analyzes a single attribute of the gametes which is more or less related to fertility [1]. In sperm analysis, simultaneously combining multiple function indicators in order to reduce the error in the fertility estimation has been largely evaluated [2]. This approach increases the accuracy of the fertility potential evaluation [3]; however, an accurate fertility estimation remains a puzzle to achieve due to the possible occurrence of unexpected variables. Nonetheless, the use of these fertility estimation tests demonstrated very reliable for excluding rather than confirming the quality of germ cells. In fact, if the lack of some gamete functionality requirements is incompatible with fertility, their presence is not a guarantee of procreative success. 

As for gametes, the functionality evaluation is also applicable to early embryos. To select embryos either in vitro produced or in vivo collected, it is necessary to identify markers disclosing maximum development competence and the birth of healthy individuals. The fields of application of this discrimination are broad and diversified according to the species. If, for example, for mammals, this selection could admit embryos to cryopreservation techniques, for marine bioindicator organisms, the evaluation of correct embryo development could be associated with dysfunctions following the exposure to various environmental stressors. In most cases, the microscopic analysis of embryo morphology is the simplest and most widely used examination because it is noninvasive, with an easy and rapid execution. This evaluation has, however, been progressively corroborated by more complex analyses to support the microscopic evaluation and/or to provide new indications that cannot be evaluated by the morphology. 

Oocytes and spermatozoa develop during gametogenesis that is underlined by meiosis, the unique process of cell division that leads to halving the number of chromosomes. The correct gamete production, maturation, activation, and interaction contribute to the gamete quality, which is an in vitro estimation of fertility [4,5]. In vitro tests have been developed with the aim of evaluating single attributes of the gametes’ functionality capable of estimating their fertility with a good degree of reliability. In spermatozoa, these tests evaluate the motility and kinetics through computerized analysis systems [6], the number and concentration [7,8], the viability [9], the morphological and ultrastructural characteristics [10], the DNA fragmentation [11,12], several biochemical activities [13], as well as the ability to interact with the female gamete (hemizona assay [14] and zona-free hamster test [15]). In mammals, the oocyte quality depends on many variables influencing the ovary, the follicle, the cumulus–oocyte complex, and, finally, the oocyte [16]. It is usually assessed by using noninvasive tests, such as phase-contrast microscopy, vital dye staining (trypan blue [17] and brilliant cresyl blue [18]), polarization light microscopy [19], as well as more sophisticated techniques, such as the genetic analysis of the polar body [20]. For embryos, quality evaluation is mostly based on morphological analysis [21,22], sometimes supported by genomics, transcriptomics, proteomics, and metabolomics from embryonic biopsies [23] or by the noninvasive profiling of the embryo culture medium secretome [24]. The development of new assessment approaches helps to deepen the analysis of the quality of gametes by obtaining complementary information capable, sometimes, of highlighting uncovered aspects of gamete and embryo functionality. The study of ionic currents through the use of electrophysiology techniques undertaken by our research group represents an alternative approach to those conventionally used to follow the dynamics of some mechanisms related to the physiology of gametes and embryos and identify associations with the criteria most commonly used to evaluate their quality. Another innovative approach developed by our research group relies on the use of fluorescent probes associated with fluorescence spectroscopy and aimed at providing detailed information on multiple metabolic and functional activities of cell populations, such as gametes, allowing reliable estimates of their quality to be obtained.

## 2. Electrophysiology

Electrophysiological techniques are tools of paramount importance for studying the role of ion channels in cells. Gametes are electrogenic cells due to the different distribution of electrical charges across their plasma membranes, known as the voltage gradient, which, in turn, generates a transmembrane resting potential (RP) [25]. In most of the cells, this is negative, ranging from −10 mV to −100 mV, and is regulated by potassium (K^+^) ions. Transient changes in ionic concentration in the cytoplasm shifts RP towards more positive values, known as plasma membrane depolarization, whereas hyperpolarization is the change of RP toward more negative values. Ion channels are proteins embedded in the gametes’ lipid bilayer that, in response to different stimuli, allow the passage of ions whose charges generate currents modifying the electrical asset of the cells. Ion channel gating may be triggered by a change in the voltage (voltage-operated channels), a ligand (second messenger-operated channels), or a mechanical stimulus (stretch-activated channels) [26]. The main ions modulating currents in biological processes are K^+^ that are more concentrated inside the cytoplasm, whereas those mostly present in the extracellular fluids are sodium (Na^+^) and calcium (Ca^2+^) cations. 

The patch clamp is the electrophysiological technique set up in 1976 that revolutionized the study of biophysics of cells and tissues. The two main configurations of the patch-clamp technique are the whole-cell and single-channel recordings (Figure 1). The first one is normally obtained after a seal between the plasma membrane and a glass microelectrode. A slight aspiration of the patched membrane allows clamping, recording of the RP, and measurement of the ion fluxes from an entire cell. The single-channel configuration, instead, without destroying the patched membrane, is able to record the ion fluxes through the channels located in the intact patched membrane. These techniques based on the application of gigaohm (GΩ)-seal resistance implemented studies of membrane electrophysiology, providing precious information about cell function and ion-mediated cellular events (for review, see [27,28,29,30]). Electrophysiological recordings applied to gametes and embryos have allowed the currents involved in many steps of the reproductive processes to be disclosed and measured, providing basic and fundamental pieces of information on the gametes’ functionality, fertilization, and embryo development in the studied species. 

### 2.1. Electrophysiological Techniques for Evaluating Gamete and Embryo Physiology

In the reproductive process, the involvement of different types of ion currents that are generated by the ion fluxes through channels located on the cell plasma membranes of either gametes or embryos in different animal species has been well demonstrated. These ion fluxes play a pivotal role in gamete maturation, activation, and reciprocal interaction and, subsequently, in the early stages of embryo development. 

Oocyte maturation involves nuclear and cytoplasmic changes. The latter is part of a complex process regulated by a series of sequential molecular events, which also involve modification of the plasma membrane permeability due to the ion current activity. Although oocyte maturation follows various patterns among species, there is a general consensus that different ion currents play a role in the resumption of meiotic maturation. Continued advancements in reproductive biology and electrophysiological applications have allowed, over the last decades, the involvement of ion channel activity in all the reproductive events to be highlighted, from gametogenesis to gamete maturation, activation, interaction, and even in the controversial process of polyspermy prevention. In embryo development, a critical role is played by the gap junctions that are peculiar channel proteins that connect blastomeres with each other, allowing the transfer of specific molecular messengers aimed at regulating the specific developmental program [25,31,32,33,34]. The main advantages and disadvantages related to the use of electrophysiology techniques are reported in Table 1.

#### 2.1.1. The Spermatozoon

During gamete activation, spermatozoa respond to messages originating from a gradient of ligands released by the oocyte extracellular coat in aquatic species [35] or the female reproductive tract in mammals. The term sperm capacitation was first suggested in the 1950s [36] and grouped all the changes that spermatozoa undergo in order to acquire the ability to fertilize the oocyte. In particular, the sequential steps of sperm activation and capacitation involve the induction of sperm motility, chemotaxis, first binding, acrosome reaction, and membrane fusion [37]. In all these events, the appropriate balance and signaling of intracellular ions in spermatozoa play a role of utmost importance (see [38] for review). As in all other cells, sperm-specific ion channels determine the inner and outer ion concentration and the permeability of the plasma membrane that result, ultimately, in establishing the RP. In mammals, during the journey from the testis to the uterus, spermatozoa encounter different ion gradients, such as Na^+^, K^+^, Cl^−^, Ca^2+^, H^+^, and HCO_3_^−^, whose concentrations have been accurately measured, determining their involvement in modulating either cell volume or RP and pH [39]. In several mammalian species, it has been well established that, during capacitation, spermatozoa undergo hyperpolarization caused by an increase in the net negative charges in the intracellular compartment. The occurrence of sperm hyperpolarization has been attributed mainly to K^+^ outward currents and consequent to the downstream activation of some messengers. In this line, SLO3 potassium channels are crucial to induce the acrosome reaction that relies on intracellular Ca^2+^ increase [40,41]. This is only one of the many examples on the functional relationship between different ion channel activities that support sperm activity and fertility potential. This evidence is further corroborated by experiments in which the inhibition or dysfunction of even only one channel type results in a reduced or impaired fertilization rate [42,43]. Until 2006, studies on the ion current activities in spermatozoa relied on the use of pharmacology and voltage- and ion-sensitive fluorescent probes, such as Ca^2+^ and H^+^ indicators that, however, gave uncertain results based on weak signals emitted [44,45]. Spermatogenetic cells are progenitors of spermatozoa that, thanks to their larger size, round shape, and immobility, offer an advantage for the application of the patch-clamp technique. Assuming that immature germ cells express the same proteins present on the mature sperm plasma membrane, studies provided some interesting cues on sperm ion channel function in gamete signaling [46]. In rat spermatogenetic cells, the presence of K^+^ and Ca^2+^ currents was first demonstrated [47] and, subsequently, these ion currents were considered responsible for a negative RP in rat spermatids [48]. 

In mouse spermatogenetic cells, pH-dependent Ca^2+^ and K^+^ currents were correlated with the state and function of mature sperm [49]. In particular, Ca^2+^ influx was suggested to be required for the onset of the acrosome reaction, whereas K^+^ conductance was correlated to the hyperpolarization and regulation of sperm fertilization potential [50,51]. The characterization of a novel chloride (Cl^−^) channel in spermatogenesis of *Caenorhabditis elegans* suggested its peculiar role in spermatid differentiation and a general role of the chloride conductance in spermatogenesis [52].

First attempts to apply direct electrophysiological recordings in sperm cells encountered serious difficulties due to the cell’s small size and volume, strong motility, and the association between the plasma membrane and some robust intracellular structures [53]. These morphological characteristics rendered it very difficult to perform a tight seal for a patch clamp between the recording pipette and the sperm plasma membrane [53,54]. Nonetheless, in 1987, a first indication of at least two channel fluxes was provided by applying the patch-clamp technique in sea urchin spermatozoa heads. However, authors reported significant experimental limitations, such as a low rate of GΩ-seal formation [55]. The patch-clamp technique applied to monolayers formed of a mixture of lipid vesicles and isolated sperm membranes had an enormous technological impact in the electrophysiological studies of the sperm cells. Despite these technical difficulties, the patch-clamp technique was applied to mouse and human spermatozoa [56,57], allowing the identification of the sperm plasma membrane region on which a tight GΩ seal could be formed with the patch pipette [53,56]. In this line, an accurate description of the selection of the electrophysiological equipment and media, sperm isolation for patch-clamp experiments, formation of the GΩ seal, the use of the whole-cell voltage clamp configuration, along with the advantages, limitations, and the most critical steps of this technology have been provided, detailed, and discussed [45,58]. In particular, in those years, a successful application of the whole-cell patch-clamp technique to completely matured human sperm allowed accurate studies of ion channels modulating sperm maturation, motility, chemotaxis, and acrosome reaction [59,60,61,62,63]. Moreover, by single-channel recording applied initially in sea urchin sperm plasma membrane, and then on spermatozoa of a series of animal models, the functional role of several types of ion channels was confirmed, including cations, such as K^+^ and Ca^2+^, and anion (Cl^−^) channels [64]. Clinical evidence also supports these indications, demonstrating that perturbation of ion channel activity due to human genetic alterations is significantly correlated with asthenozoospermia, a pathology characterized by reduced or absent sperm motility [65]. Along with motility, other steps involved in sperm activation, such as chemotaxis and acrosome reaction, are underpinned by ion channel activity. Although not all evaluated on electrophysiological recordings, the main sperm functional ion channels in the sperm physiology have been shown to be voltage-gated Ca^2+^ channels, Ca^2+^-activated Cl^−^ channels, K^+^ voltage-gated channel, voltage-gated H^+^ channels, NaV1.1–1.9, SLO3/KCNU1, which are underlined by fluxes of K^+^, Na^+^, Cl^−^, and Na^+^/H^+^ exchange, L- and T-type Ca^2+^ channels, and the members of the transient receptor potential (TRP) channel family [46,59,66,67,68,69,70,71,72]. Among the channels involved in sperm physiology and male infertility, the two most significant sperm cation channels are Catsper and Hv1, both characterized by whole-cell patch-clamp techniques. In mouse spermatozoa, electrophysiological characterization showed that Catsper is a sperm-specific, pH-sensitive Ca^2+^ channel located in the membrane of the flagellar piece and required for sperm hyperactivation [46,61,73]. A recent study utilized a technical strategy by swelling sea urchin spermatozoa and obtaining a stable cell-attached configuration. This allowed CatSper to be characterized by using the patch-clamp technique, further highlighting its role in motility in response to speract, a compound released from the egg jelly, known to induce chemotaxis [74]. Located on the flagellum midpiece of the human spermatozoa, Hv1 is a high-conductive voltage-gated proton channel that represents the main H^+^ extrusion pathway controlling sperm intracellular pH. Hv1 is activated by an alkaline environment that, by inducing intracellular alkalinization, is well recognized to promote sperm motility [75]. It has been hypothesized that there is an interplay between Hv1 and Catsper, since the latter is also potentiated by intracellular alkalinization [76]. The combined action of these two channels, together with their colocalization on the principal piece of the sperm flagellum in human spermatozoa, may be involved in increasing both intracellular Ca^2+^ and pH that are required for sperm activation in the female reproductive tract [57]. Along with this two-channel coaction, in human spermatozoa, under whole-cell recording configuration, evidence was provided of a transient inward “tail current” (ITail), whose activity is mediated by progesterone-activated distinct channels [77]. Table 2 summarizes the main research on the ion channels and ion currents in the male gamete at various stages of development and maturation.

#### 2.1.2. The Oocyte

Marine invertebrates were the most widely studied animal models for reproductive biology since the second half of the last century, and the first hints of the increase in Ca^2+^ current amplitude during oocyte germinal vesicle breakdown were provided in mollusks [78]. Later, the occurrence of Ca^2+^ flux through voltage-gated channels was confirmed in other mollusks with different maturational patterns [79,80]. The main mechanism underlying these events was spotted in an initial plasma membrane Ca^2+^ current associated with the RP depolarization that, in turn, was responsible for the mobilization of Ca^2+^ currents from the intracellular stores [81,82].

Among marine invertebrates, ascidians play a pivotal role in electrophysiological studies due to their oocyte morphological characteristics. In particular, early studies in the 1990s described the presence and modification of voltage-gated currents in either the ascidians’ mature oocyte or early embryos. Of interest were the first pieces of evidence of a random distribution of Na^+^, Ca^2+^, and K^+^ currents along the animal/vegetal axis of the oocytes blocked at the metaphase I stage. Consistent with this topographical distribution, subsequent investigations characterized two different types of voltage-dependent Ca^2+^ channels, the L- and the T-type, suggesting their specific role in the regulation of cytosolic Ca^2+^ concentration [Ca^2+^]_i_ during oocyte maturation and early embryo development [83,84,85,86,87,88,89]. Later, the pattern and the possible functional role of ion current activity have been described in either the immature oocyte stage in cephalopods or two different ascidian species [90,91]. 

Three innovative Italian studies aimed to disclose the role of ion currents in the reproductive processes of marine animal models. In the cephalopod *Octopus vulgaris*, for the first time, our team applied the whole-cell voltage clamp technique to the decorionated oocytes in the pre- and early-vitellogenic oocytes. Interestingly, we presented evidence that L-type Ca^2+^ currents together with the steady-state conductance were higher in small pre-vitellogenic oocytes and significantly lower in the larger early vitellogenic oocytes. Since these characteristics were more evident during the reproductive period, it was hypothesized that ion and L-type Ca^2+^ currents were associated with specific gamete growth stages, the vitellogenic cycle progression, and the reproductive cycle of the octopus females, suggesting also a role of these currents in preparing the plasma membrane for the imminent interaction with the spermatozoon [92]. 

In ascidians, we performed an accurate characterization along with the temporal and spatial distribution of plasma membrane voltage-dependent ion currents from the immature oocyte up to the eight-cell stage embryo in *Ciona robusta* (previously indicated as *Ciona intestinalis* spA), proposing these currents as markers of early embryogenic processes [93]. The main maturational stages used were pre-vitellogenic oocytes, which exhibited the highest L-type Ca^2+^ current activity, whereas the following vitellogenic stage showed, for the first time, the presence of Na^+^ current activity that remained high during the maturation up to the post-vitellogenic stage. At these stages, the oocyte acquires meiotic competence and the suitability to interact with the spermatozoon [94]. More specifically, by using the whole-cell voltage clamp technique, we also disclosed the ion currents involved in oocyte meiotic progression and fertilization, together with the spatial distribution of ion currents in the blastomeres at the developmental stage of eight cell (Figure 2). This stage plays a crucial relevance, as it coincides with the segregation of the different cell types’ precursors of future tissues into each blastomere [95]. In this study, we confirmed the role played by voltage-dependent Ca^2+^ currents during oocyte maturation and, in particular, for L-type calcium currents. A few years later, our group demonstrated, for the first time, the presence and the functional role of T-type Ca^2+^ currents in the growth of immature oocytes of the ascidian *Styela plicata* [96]. We classified three subtypes of immature oocytes on the basis of their size, morphology, and accessory cells. These stages were shown to be associated with increased activity of T-type Ca^2+^ currents and plasma membrane hyperpolarization. Consistently, we observed that these currents oscillated in the early embryonic stages, with an increasing amplitude starting from the zygote up to the eight-cell stage. The pharmacological inhibition of T-type Ca^2+^ currents induced a significant reduction in the cleavage rate and an absence of larval formation [96]. A wide literature reports that T-type Ca^2+^ current activities are involved in neuron, heart cell, smooth muscle cell, and endocrine cell regulation [97,98]; however, surprisingly, in this study, for the first time, we showed a novel role of these channels in modulating oocyte growth, maturation, fertilization, and embryo development in marine invertebrates. 

At fertilization, a reciprocal gamete activation occurs rescuing the metabolic quiescent gametes and making them active cells ready to fertilize and be fertilized. Starting with the oocyte-induced sperm activation and progressing toward the sperm-induced oocyte activation, a cascade of consecutive events occurs, often dependent on ion current activation [37]. Oocyte activation triggered by the interaction with the fertilizing spermatozoon is characterized by the change in electrical properties of the plasma membrane. In the echinoderms, the first studies in the 1950s demonstrated K^+^ ion fluxes through the oocyte plasma membrane that were associated with a transient change in the RP, called fertilization potential [99,100]. With the advent of the whole-cell voltage clamp technique, the fertilization potential was recorded and characterized in echinoderm oocytes by an Italian research team, showing that it was generated by the activation of a transient voltage-gated inward current [101,102]. In particular, an RP depolarization results from different ions flowing across the plasma membrane. This peculiar ion current was named fertilization current (FC) and its first electrophysiological characterization in the ascidian *Ciona robusta* demonstrated that the FC originated from the gating of high conductive and nonspecific ion channels [103,104]. However, by using an upgraded patch-clamp software, our research group showed a relevant involvement of Na^+^ currents in the FC, shedding light on the influence of the FC on subsequent embryo development [93]. Following studies confirmed that FC was responsible for the first oocyte activation events and differences in FC among species were found. FC was recorded in sea urchins and amphibians, such as Xenopus, characterizing the channels involved in FC as nonspecific and calcium-activated Cl^−^ channels, respectively [105,106]. The most interesting difference observed in FC between nonmammalian and mammalian species was that, in the former, an inward FC is accompanied by an RP depolarization, whereas, in mammals, FC is an outward current associated with an RP hyperpolarization [107]. With the exception of rabbit oocytes [108], all mammals showed consistent characteristics in the generation of FC. In fact, FC with similar biophysical characteristics were also recorded in mice, hamsters [109], and bovines [110]. In the human oocyte our team applied for the first time the whole-cell voltage clamp technique recording FC as a bell-shaped outward current accompanied by a long-lasting hyperpolarization [111]. Our further biophysical characterization of ion channels revealed that the human FC is underlined by the activity of Ca^2+^-activated K^+^ channels [112].

[Ca^2+^]_i_ rises have been described in cumulus–oocyte complexes (COCs) following luteinizing hormone (LH) exposure and candidate as a signal for resumption of meiosis in the mammalian oocyte [113,114,115]. However, the role of Ca^2+^ entry through ion channels on the plasma membrane has been described only in a few mammalian species. In the mouse, oocytes at different maturational stages were compared for the presence and activity of Ca^2+^ ions, suggesting a selective increase in the number of Ca^2+^ channels during oocyte growth which preceded nuclear maturation and were associated with the acquisition of meiotic competence [116]. Later, in the same animal model, the presence of a functional voltage-dependent Ca^2+^ channel (L-type) was described and, interestingly, a relationship between the absence and/or defects in this channel expression and the ability of oocytes to undergo the germinal vesicle breakdown that underpins the maturation onset was demonstrated [117]. In mouse oocytes, a recent study aimed to investigate the distribution pattern of different types of voltage-dependent Ca^2+^ channels and their involvement in the fertilization outcome confirmed the pivotal role of Ca^2+^ entry during mammalian fertilization and that this influx may be controlled through the N- or P/Q-type voltage-dependent Ca^2+^ channels [118].

The occurrence of ion current activity was also investigated by our group in bovine oocytes at different meiotic stages, demonstrating that the activity of L-type voltage-dependent Ca^2+^ channels was high in the immature oocytes and decreased after the breakdown of the nucleus. The concomitant decrease in the steady-state conductance from the germinal vesicle to the metaphase I and a subsequent increase in the metaphase II mature oocytes corroborated the hypothesis that the plasma membrane channels represent a suitable mode of Ca^2+^ entry into bovine oocytes during meiosis [119]. Table 3 summarizes the main research on ion channels and ion currents in the female gamete at various stages of development and maturation.

#### 2.1.3. The Embryo

Voltage-gated ion currents play a role also during embryogenesis, since, following fertilization, ion currents are generally downregulated and redistributed on the basis of specific developmental physiological needs [120]. This seems particularly true in a developing embryo, since the introduction of a new plasma membrane may modulate electrical changes exerting a profound impact on the ongoing events of growth [32,121]. First studies on this matter date back to the early 1970s, when the increase in K^+^ membrane permeability was observed during embryo development in different species, and the early cleavage divisions were associated with RP hyperpolarization [122,123]. 

The role of K^+^ conductance was analyzed in *Xenopus laevis* [124] and loach [125] cleaving embryos, where a cell cycle behavior of stretch-activated K^+^ currents was found and associated with membrane conductance and RP changes. In the following years, several studies provided pieces of evidence that cell-cycle-induced channel modulation underpins early embryogenesis [126]. 

In Xenopus, the kinetic of L-type Ca^2+^ channel subunit expression suggested its possible regulator role of Ca^2+^ influx through L-type Ca^2+^ channels in the acquisition of embryo neuronal induction [127,128,129].

Ascidians represent a good model for electrophysiological studies on embryos. A detailed description of plasma membrane electrical characteristics was performed in *Halocynthia roretzi* cleavage-arrested embryos correlating the ion current activity with the presumptive tissue regions [130]. In particular, Ca^2+^ currents were associated with the presumptive muscle blastomeres, whereas Na^+^/Ca^2+^-dependent action potentials were recorded in ectodermal cells. Interestingly, the decrease in these currents was accompanied by a gradual increase in an anomalous K^+^ current throughout the embryo development. In the early 1980s, in *Halocynthia roretzi,* neural, epidermal, and muscular tissues were related to the main ion current activities found in embryos from the 8- to 32-blastomere stage [131]. These findings suggested that blastomere membrane differentiation occurs in different steps depending on the specific tissue segregation. The evaluation of current activity pattern from the oocyte up to the eight-cell stage in the ascidian *Boltenia villosa* showed a cell cycle behavior of the ion current activity and highlighted that the Na^+^ current activity of the oocyte disappeared since the first cleavage stage. In contrast, Ca^2+^ and K^+^ currents were constantly present in blastomeres with different developmental fates, suggesting that Ca^2+^ and K^+^ channels were continuously added in new developing membranes [83]. 

During the further embryo development, the oscillatory pattern of the ion current changes suggested a key role of ion channels in the differentiation of the cell types. In fact, only muscle tissue blastomeres relied on voltage-gated Ca^2+^ currents, whereas, in the remaining blastomeres, outward K^+^ currents were required [132]. In contrast, the Na^+^ currents in oocytes were redistributed soon after fertilization and progressively decreased after the first cleavage, suggesting their specific function for fertilization and a minor role in embryogenesis [83,84,87]. 

In the ascidian *Ciona robusta*, our group performed the characterization of each blastomere plasma membrane, revealing a functional expression of Ca^2+^ and Na^+^ currents from the 2- up to eight-cell stage. We observed an oscillatory pattern underlined by a decline in Ca^2+^ and Na^+^ currents from the zygote to the four-cell stage, indicating the minor role of these currents during the first embryonic mitotic cycles. Later, a significant increase in all the current activities was observed in all the blastomeres, except for the posterior vegetal blastomere B4.1 (Figure 2) [93].

Our group demonstrated also in sea urchin embryos that the activity of different ion channels correlated with developmental stages. At the two-cell embryos, a cyclical L-type Ca^2+^ channel activity was revealed, demonstrating a specific ion channel expression and polarization in different blastomeres. The whole sea urchin embryo at the 16-cell stage includes three different size populations of blastomeres, such as macromeres, mesomeres, and micromeres. Our electrophysiological characterization demonstrated a strong polarization of Ca^2+^ current activity from the animal to the vegetal pole, with a cluster at the animal pole that gradually decreased up to disappearing in the small micromeres. However, these exhibited a higher global conductance with respect to the macromeres and mesomeres possibly driven by K^+^ currents [133]. This study provided the first hint of a functional role of Ca^2+^ channels in sea urchin embryo development, also supported by some morphological embryo anomalies observed in embryos grown in a reduced Ca^2+^ concentration [134]. 

Moreover, in mammals, a functional role of ion currents was demonstrated. In the 1980s, in mouse embryos, a biophysical characterization of the embryo plasma membrane showed the presence of inward Ca^2+^ currents progressively decreasing through early development, with the whole disappearance at the eight-cell stage. In hamsters, instead, an abrupt increase in the outward current occurred after the two-cell stage, showing an inverse correlation with the Ca^2+^ current pattern [135]. These data argued for a potential role of either Ca^2+^ or K^+^ currents during membrane differentiation in mammalian embryos.

More recent studies revealed that anion channels, such as Cl^−^ channels, also play a role in early preimplantation embryos. A change in the expression of Cl^−^ channels in preimplantation mouse embryos has been demonstrated. In particular, during in vitro embryo culture, Cl^−^ channels modulate the uptake of amino acids [136], whereas a swelling-activated anion current correlated with cell-cycle stage [137]. By using the whole-cell voltage clamp technique, the presence of a 4,4′-diisothiocyanostilbene-2,2′-disulfonic acid (DIDS)-sensitive, outwardly rectifying Cl^−^ current throughout the embryo development process was characterized, with a large conductance from the zygote up to blastocyst stages. Following blastocyst expansion, a further DIDS-insensitive Cl^−^ current was detected in cells belonging to the trophoblast lineage [138].

Gap junctions (GJ) are transmembrane ion channels that, during embryonic development, establish the inter-blastomere coupling and communication by diffusing ions and small metabolites [139,140]. GJ play a fundamental role in embryo development, such as compaction [141], cavitation [142], and embryo viability [143]. Some studies support the functional role of GJ in preimplantation development (see [144] for review); in fact, GJ are active in the early mouse embryo [145] and their perturbation at the 8–16-cell stage induces further morula decompaction and, in turn, causes developmental defects [146,147]. Electrophysiological techniques have been successfully applied in the study of GJ function during embryo development by showing the electrical communication in terms of electric charge exchange. In the ascidian *Ciona robusta*, our group demonstrated the electrical coupling between blastomeres through GJ in the two-cell embryo, together with a functional maternal expression at the zygote stage [86,148]. Similarly, in the sea urchin at the 16-cell stage, we discovered a functional GJ cluster localized at the vegetal pole exactly in the opposite side with respect to the L-type Ca^2+^ channels cluster [149]. Moreover, we showed that regionalization of GJ between macromeres and micromeres correlates with the inductive process between these blastomeres and their descendants. In support of the functional role of the electrical communication between blastomeres, it was shown that the inhibition of the GJ communication was associated with a delay in successive gastrulation and defects in archenteron formation [150]. Electrical coupling between blastomeres was also found by our group in mammals. In both in-vivo- and in-vitro-produced bovine embryos, a progressive decrease in GJ permeability was recorded during the early embryo developmental stages, together with a significant difference between these differently produced embryos. Hence, a delay and reduction in the blastomere communication occurrence discriminated the embryo quality between in vitro and in vivo embryos [151], supporting the hypothesis of a pivotal role for GJ in modulating normal embryonic development. The occurrence of GJ permeability during embryonic development appears to be stage- and species-dependent. In fact, in mice, both electrical and dye coupling are established at the eight-cell stage and are correlated with the compaction [152]. In contrast, in the human embryo, in a pioneering study, our group demonstrated that GJ coupling appears only at the blastocyst stage, possibly due to a specific requirement of blastomere communication at the crucial time of tissue polarization [153].

However, blastomere communication may also transduce specific signals. In particular, we found the occurrence of membrane serotonin receptors in the inter-blastomere contact area in the early sea urchin embryo during the cleavage division by applying a whole-cell voltage clamp. These results argued for the role of ion channel activity in the regulation of the cleavage events and a possible involvement in regulating embryo development [154,155]. Table 4 summarizes the main research on ion channels and ion currents in the early embryos at various stages of development.

## 3. Fluorescent Probes for Evaluating Cell Activities

Some features of cell functionality have been brilliantly analyzed by using fluorescent probes. These substances have the ability to establish bonds on the basis of chemical affinity with cellular structures, highlighting them and allowing their exclusive reading with the aid of detectors, such as fluorescence microscope, spectrofluorometer, or flow cytometer. For the evaluation of cellular structures and components for which it is not possible to exploit a particular chemical affinity, these probes have been linked to antibodies allowing a specific molecular identification; this technique is called immunofluorescence.

The investigation with a fluorescent microscope is an accurate method, especially if conducted with an advanced system, such as the confocal laser microscope, and constitutes an essential reference for this type of analysis, since it allows the cellular location of the fluorescent probe to be verified and monitoring of any displacement areas of the analyzed target. It is based on acquiring micrographs of the sample and quantifying the number of the cells marked or not by the fluorescent probe. The obtainable result is, therefore, the relative percentage of fluorescent cells on the total number of cells examined [156]. Using an additional image analysis, it is also possible to obtain the fluorescence intensity of the cells examined. However, this technique is time-consuming, predominantly qualitative rather than quantitative, and it is generally based on a small number of cells. 

Flow cytometry is undoubtedly a powerful and sophisticated method for rapidly analyzing large numbers of cells individually using light-scattering, fluorescence, and absorbance measurements [157]. Many cellular parameters can be determined, highlighting their distribution within the cell population. Cell characteristics, such as size, membrane potential, lipoperoxidation (LPO), intracellular pH (pH_i_), and cell content of DNA, protein, receptors, and [Ca^2+^]_i_, can be accurately evaluated [157,158]. This method is based on the passage of labeled cells through a laminar flow, passing one by one through a cell where they are illuminated by one or more lasers, the interrogation points. The scattered or emitted light is filtered by mirrors and filters, reaching several photo-detectors, where the signals are amplified. Finally, the information is digitalized and reported in fluorescent intensity units. Using a computer-assisted discrimination of labeled cells, the flow cytometer provides either the emission histogram for each single cell or the number of fluorescent cells. However, this analysis requires very expensive equipment, high professional ability, and perfectly dissociated cells [159]. 

Fluorescence spectrometry represents a reliable compromise to the above techniques because it is easy to use and capable of providing solid and repeatable results on multiple functional characteristics of the cells by using specific fluorochromes [160,161]. This method measures the total fluorescence intensity of a sample and is mostly used for free and unbound dyes in solution. Its use is much more limited on labeled suspended or adherent cells, for which it is commonly used as a derived application of the spectrofluorometer, such as the 96-well plate reader. This method does not permit a discrimination between individual cells; rather, it allows an average measurement of the fluorescence intensity representative of a cell population [162]. Weaknesses of this methodology may be attributable to the difficulty of excluding possible contaminants in the sample, such as cell debris and other components providing nonspecific fluorescence, as well as aggregates and cellular matrices potentially capable of retaining the fluorochrome and, nonspecifically, magnifying the fluorescence intensity of the sample [9,163]. This source of errors can be partially solved by combining, in the preliminary phase, this analysis with microscopic examinations of the fluorescent target and adopting appropriate positive controls.

Liquid tissues, such as blood and semen, represent excellent models for the application of all three techniques because they are based on dissociated cells. Comparisons between these techniques were carried out for the evaluation of several parameters of cellular functionality. For example, the extent of sperm DNA fragmentation analyzed by TUNEL test was evaluated by either fluorescence microscopy or flow cytometry [164], and a high correlation was found between these two methods (r = 0.720, *p* < 0.001). However, the percentage of TUNEL-positive spermatozoa assessed by flow cytometry was 2.6 times higher than that detected in optical microscopy. Comparing the mitochondrial membrane potential (MMP) of BSC-40 and HeLa G cells, either under suspension or attached status, with flow cytometry and fluorescence microplate reader (spectrofluorometry), respectively, Kalbacova and colleagues [162] showed a lower sensitivity in the latter method. Conversely, evaluating the [Ca^2+^]_i_ of indo-1-loaded A172 human glioblastoma cells stimulated by platelet-derived growth factor (PDGF), Szollosi and colleagues [165] did not find a difference in sensitivity between flow cytometry and spectrofluorometer detection. Moreover, comparing fluorescence microscopy, flow cytometry, and spectrofluorometry to quantify gene electro-transfer in suspensions of CHO and B16 cells, Marjanovic and colleagues [166] demonstrated that (i) the three techniques are highly correlated, (ii) flow cytometry measures higher values of transfection percentage compared to microscopy, and (iii) spectrofluorometry can be used as a simple and consistent method to determine gene electro-transfer efficiency on cells in a suspension. In Table 5, there is a comparison of the main advantages and disadvantages associated with the use of these three techniques based on the properties of fluorescent reports for sperm quality assessment.

### 3.1. Our Experience on Fluorescence Spectroscopy to Evaluate Gamete Functionality

Chemical, physical, and biological stresses can significantly influence the gamete quality [5]. Gamete exposure to stressful conditions can occur in the outside environment, as in aquatic organisms at spawning, or in mammals when subjected to gamete manipulations, as well as inside the body, with direct effects on gametogenesis. The studies conducted by our research team in this area covered both conditions, evaluating the effect of different stresses through multiparametric tests carried out with the prevalent use of fluorescent probes read with a spectrofluorometer. 

Heat stress (HS) is a condition normally occurring in animals associated with a significant lowering of reproductive efficiency during the summer [168] and which has, unfortunately, become an emerging threat due to global warming. The effects of HS were evaluated on sperm quality in *Mytilus galloprovincialis* [160]. Rearing sexually mature mussels within tanks at either 28 °C (HS group) or 14 °C (control group) for one month, we found in the spermatozoa of the former group: (i) a significant reduction in concentration; (ii) a biphasic pattern of motility and MMP that first increased, and then collapsed; (iii) a rapid increase in LPO up to the third week of HS; (iv) after the third week of HS, an increase in DNA fragmentation that was assessed by terminal deoxynucleotidyl transferase (TdT)-mediated dUTP nick end labeling (TUNEL) assay; and (v) a decrease in the [Ca^2+^]_i_. Ultrastructural evaluations of the spermatozoa at the transmission and scanning electron microscopy (TEM and SEM) revealed atypical morphology (i.e., sperm with a globular head, asymmetrical tail, and acrosome loss) associated with the HS exposure. MMP, LPO, DNA fragmentation, and [Ca^2+^]_i_ were evaluated by using florescent probes read with a spectrofluorometer. 

By using a similar methodology, we evaluated the effects on the ascidian *Ciona robusta* sperm quality of an emergent marine contaminant, such as nickel nanoparticles (Ni NPs) [161]. Before Ni NPs exposure, spermatozoa were loaded with different florescent probes in order to evaluate various sperm quality parameters. After 2 h of Ni NPs exposure, LPO, MMP, pH_i_, and DNA fragmentation were assessed by spectrofluorometric analysis. Moreover, aliquots of sperm suspension exposed to the same concentrations of Ni NPs were used to assess fertilizing capability and ultrastructural characteristics by TEM and SEM. Ni NPs generate oxidative stress in a dose-dependent pattern that, in turn, induced LPO and DNA fragmentation, and altered MMP and sperm morphology. Furthermore, the sperm fertilizing ability progressively decreased, whereas the incidence of anomalies in the offspring progressively increased, together with the exposure concentration of Ni NPs.

Copper oxide nanoparticles (CuO NPs) are further emerging contaminants with increasing use in industrial applications and, consequently, increasing concentration in the seawater. They could become a serious threat for reproduction and, therefore, the survival of marine animals. To evaluate the effects of this compound, sea urchin spermatozoa were exposed to increasing concentrations of CuO NPs. [169]. A panel of sperm function analyses detected by fluorescent probes and read with a spectrofluorometer has been applied together with morphological assessment by SEM. Results showed that CuO NPs exposure decreased sperm viability, impaired mitochondrial activity, and increased the production of ROS, LPO, DNA damage, and morphological alterations. By verifying that the effects associated with CuO NPs were avoided following the use of antioxidants, we hypothesized that oxidative stress is the main driver of CuO NP spermiotoxic effects. 

Another physical stress directly associated with global climate change and, more precisely, with the increase in carbon dioxide (CO_2_) levels in our atmosphere is ocean acidification. Experiments were carried out to evaluate short-term (7-d) effects deriving from the direct exposure of sexually mature individuals of the ascidian *Ciona robusta* to seawater pH conditions simulating the ocean conditions predicted for the end of this century (pH = 7.80 vs. pH = 8.20 in controls), either in in situ transplant experiments at a naturally acidified volcanic vent area or in microcosm experiments [170]. Sperm parameters, such as motility, viability, MMP, LPO, intracellular and extracellular pH, intracellular levels of hydrogen peroxides and superoxide anions, as indicators of reactive oxygen species (ROS), as well as fertilization capability and morphological characteristics at SEM, were daily evaluated. In the first days of exposure to acidified conditions, sperm motility, morphology, and physiology were significantly altered. However, in the next days, there was a rapid recovery of physiological conditions, suggesting a resilience ability of ascidian spermatozoa in response to ocean acidification. A similar study was carried out in *Mytilus galloprovincialis* either in in situ transplant or microcosm experiments on a parental longer (21-d) exposure to low pH conditions [171]. Under field conditions, the low pH exposure was associated with an increase in total sperm motility and MMP on days 7 and 14, whereas, in microcosm, an increase in total motility was only found on day 14. Sperm morphology and intracellular pH were affected in both experimental approaches; however, oxidative stress was detected only in spermatozoa collected from mussels under field conditions, suggesting that multi-stress conditions may have occurred under field conditions. Altogether, these results seem to exclude the involvement of oxidative stress in ocean-acidification-related mechanisms acting on marine organisms’ reproductive failures. This topic was further deepened by direct in vitro exposure of mussel and ascidian spermatozoa to low pH conditions [172]. The pH lowering as a possible source of stress is a concrete threat, since, in free-spawning marine invertebrates, fertilization is highly sensitive to changes in seawater quality and chemistry [173]. Analyzing several endpoints of sperm functionality, such as motility, vitality, MMP, oxidative state, and pH_i_, we found that, following sperm in vitro exposure to acidified seawater, the percentage of motile spermatozoa, mitochondrial activity, and pH_i_ decreased in comparison to control, whereas vitality and oxidative state were unaffected by the low external pH in both the species. The lack of sperm activation naturally occurring at spawning by the alkaline pH of seawater would be at the basis of the lowering of sperm motility; this occurrence is strongly associated with a lowering of fertility leading to relevant implications for the fitness and the survival of marine invertebrates.

Other possible sources of stress affecting the gamete quality are related to micromanipulation and in vitro culture techniques. On this line, we analyzed bioenergetics, kinetics, and oxidative status in the semen of donkey stallions, which was collected, diluted with different extenders at different sperm concentrations, and stored at +4 °C [174]. The storage produced a progressive decline in sperm kinetics and MMP, whereas parameters related to oxidative status either increased (LPO and nitroblue tetrazolium (NBT) assay) or decreased as the antioxidant activity evaluated by anti-LPO. The anti-LPO potential was assessed by incubating C11 BODIPY^581/591^-loaded sperm with a mild oxidant stimulus; this index showed the highest correlations with sperm motility and kinetics. Extenders and sperm concentration proved to be differently effective in preserving sperm kinetics, MMP, and oxidative status. The storing-dependent sperm quality decrease has also been studied in cattle, in which frozen/thawed sperm was incubated with substances capable of stimulating sperm metabolic activity and analyzed in relation to bioenergetics, kinetics, and oxidative status [175]. Comparing several compounds, such as myo-inositol, pentoxifylline, penicillamine + hypotaurine + epinephrine mixture (PHE), caffeine, and coenzyme Q10 + zinc + D-aspartate mixture (CZA), we found that, on the first hour of incubation, CZA treatment produced the best performance in total and progressive sperm motility, as well as in sperm kinematic parameters. MMP showed the highest values after treatment with pentoxifylline and PHE. LPO and [Ca^2+^]_i_ were significantly affected by the incubation time rather than the treatments, whereas pH_i_ varied significantly in relation to either the incubation time or treatments. Significant correlations were found between sperm kinetic and metabolic parameters. 

Focusing on the high sperm sensitivity to oxidative stress and on the necessary availability of cysteines for the synthesis of the main ROS scavengers present in the genital tract, such as hypotaurine and glutathione, we have undertaken a study on the role of the 1-carbon cycle (1-CC) on sperm quality [176]. Human, bovine, and ascidian spermatozoa were incubated with a mixture of compounds supporting the 1-CC (TRT) and compared to the effects induced solely by N-acetyl-cysteine (NAC). After 90 and 180 min of incubation, the MMP in TRT and NAC groups was significantly higher than in control. However, ROS production differed between species and, only in mammalian sperm, both the 1-CC supporting mixture and NAC improved sperm kinetics, MMP, and ROS production. Conversely, in ascidian sperm, the treatment supporting I-CC failed its scope. 

The use of fluorescent dyes read with a spectrofluorometer was also applied to assess the quality of female gametes. In mammals, however, unlike spermatozoa, the quantity of oocytes available for such examinations is extremely limited. An indirect evaluation study was, therefore, undertaken through the analysis of granulosa cells [177]. Mitochondrial activity, evaluated by steroidogenic acute regulatory (StAR) protein expression and MMP, has been assessed in bovine granulosa cells (GCs) and related to follicle growth and atresia. Atresia was estimated by morphological examination of follicle walls and cumulus–oocyte complexes [178] and assessed by TUNEL assay. StAR protein expression was assessed using both immunofluorescence (IF) and Western blot (WB) analyses. GCs collected from small nonatretic follicles showed a higher MMP and WB-based StAR protein expression than small atretic follicles. For IF analysis, StAR protein expression in large atretic follicles was higher than that in large nonatretic follicles. These results suggest a role played by mitochondria in GC steroidogenic activity, which declines in healthy follicles along with their growth. In large follicles, steroidogenic activity increases with atresia. 

All the research mentioned above has progressively confirmed the effectiveness of the techniques based on the use of fluorescent probes read with a spectrofluorometer as an excellent diagnostic investigation tool to provide information on the function of the gametes under real-time and vital conditions. Some of these techniques were simultaneously tested in the gametes of several animal species, comparing their versatility. For example, the gamete viability test based on the dual DNA fluorescent dyes, i.e., SYBR-14 staining live and propidium iodide staining degenerated/dead spermatozoa, was used to evaluate the sperm viability in three marine invertebrates: the ascidian *Ciona intestinalis*, the sea urchin *Paracentrotus lividus*, and the mollusk *Mytilus galloprovincialis* [179]. This method proved effective in all three animal species and was proposed as a simple, accurate, rapid, sensitive, and cost-effective method for screening marine pollutants for spermiotoxicity.

### 3.2. The Reliability of the Fluorescence Spectroscopy Techniques

The above studies allowed a panel test to be set up capable of evaluating different characteristics of sperm function. Some of these evaluations, such as those associated with mitochondrial activity or the oxidative state, have been related to each other to derive possible associations [180]. However, even when some parameters are highly correlated with each other and their simultaneous use could be questioned for an obvious analytical repetition, they may provide useful information on the dynamics of a given stressful event. For example, LPO could be considered an obvious effect of an increase in ROS production and is usually highly correlated with the ROS content in the cells examined [180]. However, it should be considered that LPO occurs more slowly than sudden increases in ROS; moreover, it could be generated by extracellular stimuli and/or due to the reduction in antioxidant substances or scavengers. 

To verify the effectiveness of these methods for the evaluation of the functional cell characteristics, before being used for analytical purposes and read with a spectrofluorometer, each dye was incubated with the cell target and analyzed with confocal laser microscopy, to confirm the specific intracellular localization of the probe (Figure 3). In the case of spermatozoa, a highly diluted paraformaldehyde solution was used to induce a reversible sperm immobilization while maintaining vitality [181] and, thus, allowing effective traceability of the fluorescent probe and acquiring images.

Finally, each test was supported by the use of positive controls aimed at increasing or reducing the functionality of the dye target with the detection of the variation in fluorescence intensity. For this purpose, the analysis aimed at evaluating the MMP, based on JC-1 dye, used the carbonyl cyanide 3-chlorophenylhydrazone (CCCP), a protonophore that, uncoupling the oxidative phosphorylation in mitochondria, causes a decrease in MMP, shifting down the second fluorescent emission peak at ~595 nm [160]. For LPO detection based on the C11-BODIPY^581/591^ lipophilic fluorophore, the samples used as positive controls are incubated with FeSO_4_ and ascorbic acid [182]; this treatment causes an increase in the first fluorescent emission peak at ~520 nm and a decrease in the second fluorescent emission peak at ~590 nm [160]. For ROS detection, positive controls are obtained by incubating cell samples with either oxygen peroxide or pyrogallol/menadione when 2′,7-dichlorodihydrofluorescein diacetate (H_2_DCF-DA) or dihydroethidium (DHE) dyes are used [176,180]; with both treatments, the fluorescent emission peak at ~530 and ~600 nm, respectively, largely increased. For [Ca^2+^]_i_ evaluated by Fluo-4 AM dye, a very rapid incubation with a Ca^2+^ ionophore, such as A23187, represents an efficient positive control [160]. For DNA fragmentation, as recommended by TUNEL kit manufacturer, positive-control samples are treated with DNase I [160]. For cell viability based on SYBR-14-PI assay, positive controls have been prepared in various ways; however, particularly suitable for marine invertebrate sperm is heating the sample to 50 °C for 15 min before using the dyes [179]. The pH_i_ evaluation is carried out using BCECF-AM fluorescence probe and based on the intensity of the fluorescence emission at 530 nm induced by excitation with two different wavelengths (440 and 490 nm). The transformation of the fluorescence intensity value into the pH value is obtained thanks to a calibration line. This derives from the exposure of cellular samples in a culture medium without buffer and in the presence of nigericin, which annuls the ability of the cells to regulate their intracellular pH, making it equal to the external one. By setting the sample solutions at different pH values and evaluating the fluorescence intensity of these cells, a straight line is obtained by whose equation the transformation of the data is obtained. The effectiveness of this method is confirmed by a high correlation coefficient of the calibration line [160]. 

## 4. Conclusions

In this review, two techniques widely used by our research group to evaluate the characteristics of the biology and functionality of gametes and embryos, ranging from marine invertebrates to humans, were evaluated in detail. Electrophysiology techniques undoubtedly represent sophisticated methods, difficult to apply in the diagnostic routine but capable of uncovering fine dynamics of the mechanisms of gamete maturation, fertilization, and embryonic development. The fluorescence spectrometry techniques, on the other hand, have been conceived as simple and fast techniques that are, therefore, easy to apply and able to provide multiple details on the functioning and quality of the cells examined. We believe that, through the development of studies based on the application of these methods, important information has been, and we hope will be, revealed on the mechanisms of gamete function and embryo development.

## Figures and Tables

**Figure 1 biomolecules-12-01685-f001:**
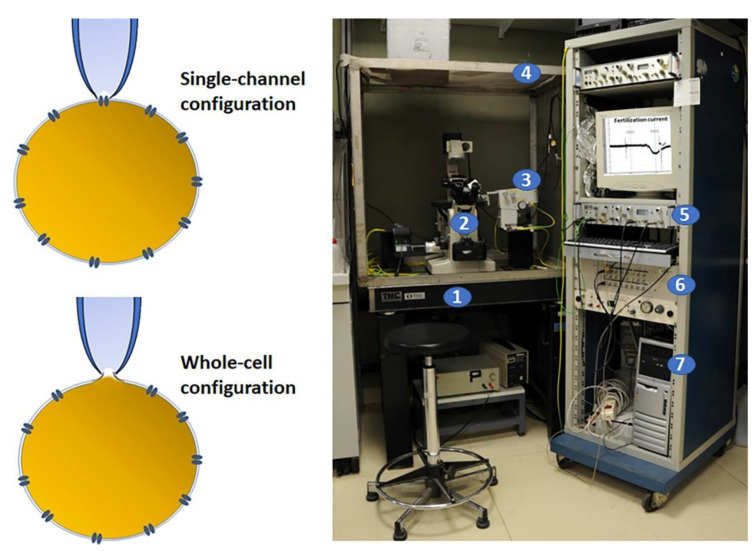
Electrophysiology patch clamp and set up. On the left, a schematic diagram of two recording methods for patch clamp. This technique requires the initial formation of a tight seal between the plasma membrane and the blunt tip of a glass micropipette that is obtained by applying a light suction. In this configuration, the currents flux through the channel into the pipette and can be recorded by an electrode connected to a highly sensitive differential amplifier (single channel) or, by applying a stronger suction, the membrane patch can be ruptured and the interior of the pipette becomes continuous with the cytoplasm, allowing the electrical potentials and currents from the entire cell (whole cell) to be recorded. On the right, the electrophysiology set up that includes: (1) an antivibration air table to minimize mechanical vibrations; (2) an inverted microscope; (3) a micromanipulator for stably positioning the micropipette; (4) a Faraday cage to block external electrical interference; (5) an amplifier to collect and amplify the electrical signals; (6) a digitizer to convert analogue into digital signals; (7) a computer with a proper analytical software. On the PC screen, there is a typical fertilization current recorded in ascidian oocytes.

**Figure 2 biomolecules-12-01685-f002:**
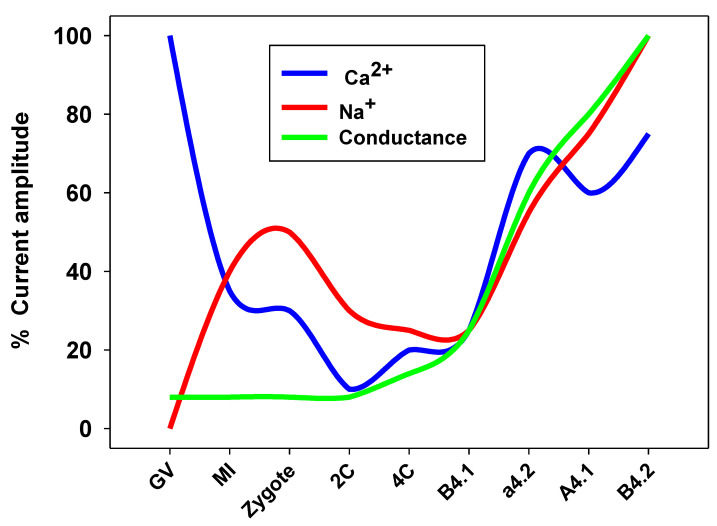
Ion currents in *Ciona robusta* oocytes and embryo in different developmental stages. GV (germinal vesicle, immature oocyte), MI (metaphase I stage oocyte), 2C (2-cell stage), 4C (4-cell stage). B4.1, a4.2, A4.1, and B4.2 are blastomeres belonging to the 8-cell stage embryo.

**Figure 3 biomolecules-12-01685-f003:**
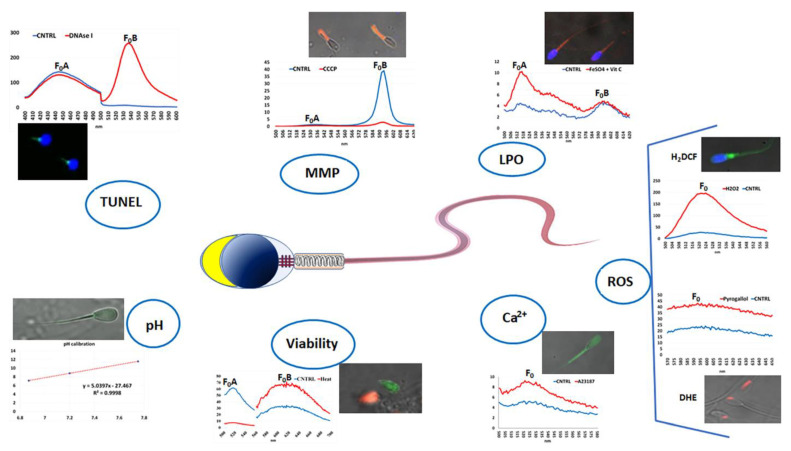
Schematic representation of the fluorescence spectra and photomicrographs of spermatozoa of different species stained with different fluorescent probes, evaluated by confocal laser microscopy and assessed by fluorescence spectrometry. DNA fragmentation was assessed in mussel sperm by using TUNEL assay and DAPI and comparing the two fluorescent emission peaks; positive controls were obtained by treating sperm with DNAse I. Mitochondrial membrane potential (MMP) was assessed in donkey sperm by JC-1 staining and comparing the two fluorescent emission peaks (FoB/FoB); positive control was obtained by incubating sperm with CCCP. Lipoperoxidation (LPO) was assessed in donkey sperm by C11-BODIPY^581/591^ staining and comparing the two fluorescent emission peaks ((F0A/(FoA + FoB)) × 100); positive control was obtained by incubating sperm with FeSO_4_ and vitamin C. Reactive oxygen species (ROS) were assessed in bovine sperm by either H_2_DCF-DA or DHE for evaluating the hydrogen peroxide or superoxide ion content, respectively; positive control samples were obtained by treating sperm with hydrogen peroxide and pyrogallol, respectively. Intracellular calcium content [Ca^2+^]_i_ was assessed in bovine sperm by Fluo-4 AM staining; positive controls were obtained by treating sperm with the calcium ionophore, A23178. Viability in sea urchin sperm was assessed by SYBR-14/PI staining; positive controls were obtaining by treating sperm with heat. Intracellular pH was assessed in bovine sperm by using BCECF-AM; a high correlation coefficient (R^2^) of the three measurements used to obtain the calibration line ensures the reliability of the method. Fo indicates the emission peak of the fluorescence intensity.

**Table 1 biomolecules-12-01685-t001:** Advantages and disadvantages associated with the application of the electrophysiological technique in gametes and early embryos.

Advantages (Pros)	Disadvantages (Cons)
Real-time evaluation of the channel activity Clear and reproducible results Objective evaluation Low maintenance costs	High cost of the equipment High professional skills Difficulties in membrane cell manipulation (fragile and sticky membranes) Need to remove extracellular coats Need immotile cells

**Table 2 biomolecules-12-01685-t002:** Types and functions of ion channels/currents involved in animal and human spermatogenesis.

Animal Species	Developmental Stage	Channels/Currents	Functions	References
*C. elegans*	Spermatids	Cl^−^	Differentiation	[52]
Marine invertebrates	Spermatozoa	T-type Ca^2+^, Ca^2+^-activated Cl^−^ channels, K^+^, TRP family	Chemotaxis, motility	[46,66,72]
Mammals	Several spermatogenetic stages	Ca^2+^-activated Cl^−^ channels, K^+^, H^+^, NaV1.1–1.9, SLO3, L- and T-type Ca^2+^, TRP family	Development, maturation, chemotaxis, motility, capacitation, acrosome reaction	[46,59,66,67,69,70,71,72]
Rat	Spermatocytes	K^+^, Ca^2+^	Maturation	[47,48]
Rat, human	Epididymal and ejaculated sperm	K^+^, Ca^2+^, Cl^−^	Sperm physiology, gamete interaction	[64]
Mouse	Spermatozoa	SLO3	Acrosome reaction	[40,41]
	Spermatocytes	K^+^, Ca^2+^	Maturation, acrosome reaction, fertilization potential	[49,50]
	Spermatozoa	CatSper, Hv1	Motility	[74]
Human	Spermatozoa	K^+^, Cl^−^, Ca^2+^, K^+^, HCO_3_^−^, Na^+^	Maturation, motility, chemotaxis, acrosome reaction	[59,60,61,62,63]
	Spermatozoa	HV1	Motility, pH control	[53]
	Spermatozoa	ITail	“Tail current” or membrane repolarization	[77]

**Table 3 biomolecules-12-01685-t003:** Types and functions of ion channels/currents involved in oogenesis and oocyte maturation in animals and humans.

Animal Species	Developmental Stage	Channels/Currents	Functions	References
*Ciona intestinalis* (now *Ciona robusta)*	Metaphase I	Na^+^, Ca^2+^, K^+^ L- and T-type Ca^2+^	Animal/vegetal axis establishment Oocyte maturation and early embryo development	[85,86,87,88,89,90,91] [90,91]
	Pre-vitellogenic vitellogenic	L-type Ca^2+^ Na^+^	Maturation Sperm interaction	[94]
	Metaphase I	fertilization current; Na^+^	Fertilization	[93,103,104]
*Styela plicata*	Germinal vescicle	T-type Ca^2+^	Growth, maturation, fertilization	[96]
Echinoderms	Mature egg	K^+^	Fertilization	[99,100]
Sea urchin	Mature egg	fertilization current	Fertilization	[101,102]
	Mature egg	Non specific	Fertilization	[105]
*Octopus vulgaris*	Pre-vitellogenic and early vitellogenic	L-type Ca^2+^	Vitellogenic cycle progression	[92]
*Xenopus laevis*	Mature egg	Ca^2+^-activated Cl^−^	Fertilization	[106]
Rabbit	Metaphase II	Depolarizing	Fertilization	[108]
Hamster, mouse	Metaphase II	Hyperpolarizing	Fertilization	[109]
Mouse	Cumulus–oocyte complexes	Ca^2+^	Growth and meiotic competence	[113,114,115,116]
	Germinal vesicle	L-type Ca^2+^	GVBD and growth	[117]
	Metaphase II	N- or P/Q-type Ca^2+^	Fertilization	[118]
Bovine	Metaphase II	Ca^2+^- activated K^+^	Fertilization	[110]
	Germinal vesicle	L-type Ca^2+^	Meiosis resumption	[119]
Human	Metaphase II	Ca^2+^-activated K^+^	Fertilization	[111,112]

**Table 4 biomolecules-12-01685-t004:** Types and functions of ion channels/currents involved in early embryo development in animals and humans.

Animal Species	Developmental Stage	Channels/Currents	Functions	References
*Ciona intestinalis* (now	2-cell and zygote	Gap junction activity and expression	Developmental competence	[86,148]
*Ciona robusta*)	2- up to 8-cell	Ca^2+^ and Na^+^ oscillatory pattern	Blastomere development	[93]
*Boltenia villosa*	8-cell Post- gastrulation	Ca^2+^, K^+^, Na^+^ voltage-dependent Ca^2+^	Membrane development Muscle-tissue development	[83,84,87] [132]
*Halocynthia roretzi*	Cleavage-arrested embryos	Ca^2+^ Na^+^/Ca^2+^-dependent action potentials	Specific tissue segregation	[130]
	8- up to 32-cell	Unspecific	Neural, epidermal, and muscular tissue segregation	[131]
Starfish	Germinal vesicle	K^+^	Membrane hyperpolarization	[122]
Sea urchin	2-cell Early embryo	L-type Ca^2+^ L-type Ca^2+^	Cell-cycle-related Development	[133] [134]
	16-cell	Gap junctions	Gastrulation and archenteron formation	[150]
	Cleavage	Unspecific	Development regulation	[154,155]
*Xenopus laevis*, loach	Cleaving embryos	stretch-activated K^+^	Cell-cycle-induced channel modulation	[124,125]
*Xenopus laevis*	Mature eggs	L-type Ca^2+^	Neural induction	[127,128,129]
Mouse, hamster	2- up to 8-cell	Ca^2+^, K^+^	Membrane differentiation	[135]
Mouse	Preimplantation embryos	Cl^−^	Amino acids uptake Cell cycle	[136,137]
	Zygote to blastocyst	DIDS-insensitive Cl^−^	Blastocyst expansion, trophoblast lineage	[138]
	Early embryo	Gap junctions	Embryo compaction	[145,146,147]
Bovine	In vivo and in vitro embryos	Gap junctions	Embryo quality modulation	[151]
Human	Blastocyst	Gap junctions	Tissue polarization	[153]

**Table 5 biomolecules-12-01685-t005:** Advantages and disadvantages in the use of fluorescence microscopy, flow cytometry, and fluorescence spectrometry techniques for the evaluation of sperm functional parameters.

	Advantages (Pros)	Disadvantages (Cons)
Fluorescence microscopy	Low cost of the equipment Localization of the dye Visual quality check Morphological info High accuracy for qualitative assessments Multiple ways of discriminating mixed cell populations	Time consuming Small reference cell sample Subjective analysis (operator effect) High professional skills Moderate accuracy for quantitative assessments
Flow cytometry or fluorescence-activated cell sorting (FACS)	Rapid assessment of large cell populations High accuracy Objective analysis Sorting cell populations Excluding unclear signals Multiple ways of discriminating mixed cell populations	High cost of the equipment Long test times Needs visual quality check first * Less morphological info * High professional skills
Fluorescence spectrometry	Low cost of the equipment Short test times Rapid assessment of large cell populations Objective analysis Moderate professional skills	Needs visual quality check first Less morphological info Difficulty to exclude unclear signals (cell debris, dye aggregates) Applicable on single or fluorescence-discriminable cell populations

* These disadvantages have now been solved with image flow cytometry and its upgrades, as virtual-freezing fluorescence imaging flow cytometry [167].

## Data Availability

Not applicable.

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
