# Peer review of "Electrophysiology and Fluorescence Spectroscopy Approaches for Evaluating Gamete and Embryo Functionality in Animals and Humans"

_biomolecules, 2022, doi:10.3390/biom12111685_

Round 1
Reviewer 1 Report
In this review, the authors summarize work done by them and other teams to assess gamete and embryo functionality in different animal species, using electrophysiology and fluorescence spectrometry. The review is well documented and comprehensive, but lacks tables and diagrams to help the reader follow. For example, for section 2., a diagram describing the experimental setup for whole-cell and single-channel electrophysiology would be helpful, as well as the analysis pipeline. Also missing are three tables summarizing what the electrophysiology has led to the understanding of the quality of the 1/sperm (section 2.1), 2/oocyte (section 2.2), 3/embryo (section 2.3) as a function of developmental stages and species, which would allow for a direct comparison between processes and species. For section 3., a comparative table of the 3 techniques (investigation with a fluorescent microscope, flow cytometry, fluorescence spectrometry) with the advantages and disadvantages would be a plus.
Author Response
We thank the reviewer a lot for the valuable suggestions that have allowed us to improve our paper significantly. We have constructively accepted all the information provided, so we have included 1 new Figure and 5 new Tables in this review. In Figure 1, we described the experimental setup for whole-cell and single-channel electrophysiology and the instrumentation dedicated to this analysis. In Tables 1 and 5, we have listed the advantages and disadvantages related to the use of electrophysiology techniques and the three techniques examined (fluorescent microscope, flow cytometry, fluorescence spectrometry) based on the use of fluorescent probes. At the end of each sub-chapter of Section 2, we inserted a table summarizing the main discoveries/descriptions of ionic currents and channels highlighted in the gametes (Tables 2 and 3) and in the early stages of embryonic development (Table 4).
Reviewer 2 Report
In this review, the authors summarized two techniques in their lab for evaluation of gametes and embryos. Overall, it is a well-written and clear review article. And, the research area would be interesting to readers. However, I still have some comments as follows:
1. Whilst it is no doubt that the authors’ group did a lot of work and had many publications in this research field, I would recommend to summarize and categorize the techniques in this field firstly, and then introduce each technique to give the readers a clear view of all the techniques.
2. I would like to encourage the authors to include more others’ work to strengthen their opinions.
3. What are the relationship, advantages and disadvantages, and applications of these techniques?
Author Response
We thank the reviewer a lot for the valuable suggestions that have allowed us to improve our paper significantly. We have constructively accepted all the information provided, so we have included 1 new Figure and 5 new Tables in this review. In Figure 1, we described the experimental setup for whole-cell and single-channel electrophysiology and the instrumentation dedicated to this analysis. In Tables 1 and 5, we have listed the advantages and disadvantages related to the use of electrophysiology techniques and the three techniques examined (fluorescent microscope, flow cytometry, fluorescence spectrometry) based on the use of fluorescent probes. At the end of each sub-chapter of Section 2, we inserted a table summarizing the main discoveries/descriptions of ionic currents and channels highlighted in the gametes (Tables 2 and 3) and in the early stages of embryonic development (Table 4). Regarding the possibility of including others' work to strengthen our findings, we understand the perplexities of the reviewer; however, this paper is part of a special issue entitled “State-of-the-Art Molecular Reproduction in Italy”. Based on this title, we have built a paper that summarizes our research using as guiding threads two methodological approaches that we have widely used, i.e., electrophysiology and fluorescence spectrometry. While for electrophysiology, the technique was used by a few other research groups that we have extensively cited, for fluorescence spectroscopy we have not found many other studies in the literature that used this methodological approach for evaluating gamete function. Furthermore, taking into account the 182 papers cited, 48 of them refer to members of our research group and 134 are attributable to other research groups.
Round 2
Reviewer 2 Report
No other comments.